# Cage or Pelvic Graft—Study on Bony Fusion of the Ventral Thoracic and Lumbar Spine in Traumatic Vertebral Fractures

**DOI:** 10.3390/medicina57080786

**Published:** 2021-07-31

**Authors:** Katharina Jäckle, Theresa Brix, Swantje Oberthür, Paul Jonathan Roch, Stephan Sehmisch, Wolfgang Lehmann, Lukas Weiser

**Affiliations:** Department of Trauma Surgery, Orthopaedics and Plastic Surgery, University Medical Center Göttingen, Robert-Koch Str. 40, 37075 Göttingen, Germany; katharina.jaeckle@med.uni-goettingen.de (K.J.); Theresa.Brix@szb-chb.ch (T.B.); swantje.oberthuer@med.uni-goettingen.de (S.O.); jonathan.roch@med.uni-goettingen.de (P.J.R.); stephan.sehmisch@med.uni-goettingen.de (S.S.); Wolfgang.Lehmann@med.uni-goettingen.de (W.L.)

**Keywords:** bony fusion, spine surgery, cage implantation, autologous pelvic bone graft

## Abstract

*Background and Objectives:* Stabilization of the spine by cage implantation or autologous pelvic bone graft are surgical methods for the treatment of traumatic spine fractures. These methods serve to stably re-adjust the spine and to prevent late detrimental effects such as pain or increasing kyphosis. They both involve ventral interventions using interbody fusion to replace the intervertebral disc space between the vertebral bodies either by cages or autologous pelvic bone grafts. We examined which of these methods serves the patients better in terms of bone fusion and the long-term clinical outcome. *Materials and Methods:* Forty-six patients with traumatic fractures (12 cages; mean age: 54.08/34 pelvic bone grafts; mean age: 42.18) who received an anterior fusion in the thoracic or lumbar spine were included in the study. Postoperative X-ray images were evaluated, and fusion of the stabilized segment was inspected by two experienced spine surgeons. The time to discharge from hospital and gender differences were evaluated. *Results:* There was a significant difference of the bone fusion rate of patients with autologous pelvic bone grafts in favor of cage implantation (*p* = 0.0216). Also, the stationary phase of patients who received cage implantations was clearly shorter (17.50 days vs. 23.85 days; *p* = 0.0089). In addition, we observed a significant gender difference with respect to the bony fusion rate in favor of females treated with cage implantations (*p* < 0.0001). *Conclusions:* Cage implantations after spinal fractures result in better bony fusion rates as compared to autologous pelvic bone grafts and a shorter stay of the patients in the hospital. Thus, we conclude that cage implantations rather than autologous pelvic bone grafts should be the preferred surgical treatment for stabilizing the spine after fracture.

## 1. Introduction

The incidence of vertebral body fractures is age-dependent. They affect only about 5.7/1000 among people under 60 years but increase significantly with age [1,2]. In patients who are younger than 60 years, men are twice as likely to be affected as women, whereas patients older than 60 years are more likely to be female [1,2]. The reason for the age-related increase in female patients is the dominance of osteoporotic vertebral body fractures. In general, the incidence of vertebral body fractures, which has been steadily increasing in recent years, correlates with leisure activities in the high-friction area and increasing mobility in old age [3]. The main causes of injury are traffic accidents and falls from higher altitudes [4]. While fusion of the ventral spine is regularly performed in patients with degenerative diseases, patients with a traumatic vertebral body fracture only rarely need a ventral fusion.

From a biomechanical point of view, ventral stabilization in the case of burst fractures of the spine has the advantage that the therapy starts directly at the destroyed ventral column [5]. Numerous publications have shown that the sole dorsal instrumentation of type A4, B, and C injuries of the thoracolumbar transition is associated with a secondary loss of correction [4,5]. The destruction of the adjacent intervertebral discs, especially in the event of a herniation of nucleus tissue in the injured vertebral body, means that they are no longer able to maintain the statics of the movement segment after removal or if the dorsal implants fail [4]. The main indications for ventral stabilization are an expected loss of correction after a pronounced bony defect and after corpectomy if the intervertebral disc shows significant damage [4]. One of the several methods uses synthetic material such as metals and ceramics or involves autologous bone grafts to replace the intervertebral disc between the vertebral bodies. Cage insertion serves as a placeholder for the removed intervertebral discs. The use of cage systems is therefore a method to fix instable vertebral bodies caused by fractures, malignant infestation and/or to preventively clear out potentially unstable spinal areas. Instabilities due to vertebral body destruction, irrespective of their cause, are per se indications for surgery [6]. As an alternative to cage implantations, discs can be replaced by autologous pelvic bone grafts, a long-established method [7,8,9].

Both cage insertions and pelvic bone grafts can serve as a monosegmental placeholder. The bone can subsequently grow through the disc space to ensure a bony fusion of the adjacent vertebrae. This fusion eventually stabilizes the affected spinal segment, which is thereby fixed in the corrected, proper anatomical position. This effect is mostly supported by insertion of a fixateur interne from dorsal to additionally stabilize the anteriorly inserted implant [4]. With a few exceptions in the area of the upper cervical spine and lower spine, ventral care is always combined with dorsal stabilization [4].

## 2. Materials and Methods

### 2.1. The Patient Collective for Study

The present study was approved by the ethics committee of the University hospital (approval number: AN 3/7/19) and was in compliance with the Helsinki Declaration.

Inclusion criteria for the reported retrospective cohort study with observing character were defined by a patient collective who received a monosegmental fusion at the thoracic and lumbar spine due to a traumatic injury and with complete follow up records over a minimum of one year. The study was carried out at The University Medical Center in the period 2008–2018 with thoracic and lumbar spine surgery for patients with cage implantation (see Figure 1a,b) and for a six year period (2008–2013) for a patient collective with autologous tricortical pelvic bone grafts (see Figure 1c,d) with regular follow-up examinations after the surgery procedure. Based on these criteria, a patients collective treated with monosegmental cages, those with a traumatic fracture in the area of the thoracic vertebrae and lumbar vertebrae, were selected. They had been subjected to a ventral monosegmental fusion by rather a cage (Monolift^XP^ TL^®^, Aesculap AG, Tuttlingen, Germany) or pelvic bone graft. Of these 103 patients, 12 patients (mean age: 54.08; 5 females and 7 males) treated with cages were included in the study, as were 34 patients (mean age: 42.18; 11 females and 23 males) out of the 95 patients treated with autologous pelvic bone grafts.

Exclusion criteria were follow-up examinations less than one year and pathological or degenerative fractures. Patients were recalled at 3 months and 12 months for a standard follow-up. However, in some cases the patients were not present exactly 12 months after the surgery and thus, the control examination took place a few months later or, in a few cases, years later.

Since standard classifications on the bony fusion of caging and autologous pelvic bone graft are not yet established, we assessed the approximate fusion status via X-ray analysis. These radiological controls included an antero-posterior view of the spine with an additional lateral projection and they were evaluated by two experienced spine surgeons. The status of the patients was considered stable, and bones were judged as probably fused, when the following radiological signs were detectable: (1) loosening signs such as hypodense areas around the implant, bone lysis, hypertrophic callus, or delayed fracture healing were absent after 12 months [10]; (2) detectable callus formation when bridging bone connecting the adjacent vertebral bodies was detected either through or around the implants and no radiolucency was seen, (3) visible bridging trabecular bone either crossing the cage or surrounding it, which were detected on anterior–posterior and lateral views of the radiographs [11]; (4) lack of substantial sclerotic changes in the recipient bone bed; and (5) vertebral body translation of <3 mm on lateral radiographs [11]. If these radiological signs were absent, it was classified as non-fused (see Figure 1e,f).

### 2.2. Surgical Intervention

All patients involved in the study with cage implantation (see Table 1) or autologous pelvic bone graft (see Table 1) were undergoing a ventral spondylodesis as the operational treatment as part of a dorsoventral spinal fusion. The intervertebral disc is removed between two vertebrae and the resulting gap is either filled with a poly-ether-ether-ketone (PEEK) tantal coated cage or autologous tricortical pelvic bone graft. In all cases the vertebrae were initially stabilized with a fixateur interne from the dorsal side. The subsequent ventral stabilization was usually carried out during another inpatient stay at an average of approximately 3.69 months (±4.14 months), depending on the patient’s state of health.

The dorsal stabilization was carried out with the krypton^®^ system (Ulrich Medical GmbH & Co. KG, Ulm, Germany) or URS system (DePuy Synthes Company, Raynham, MA, USA). During the stabilization operation in the area of the lumbar spine, a vertebral body above and a vertebral body below the inserted cage/autologous tricortical pelvic bone grafts were stabilized with each other by screwing in pedicle screws and inserting longitudinal connector rods. In the area of the thoracic spine, two vertebral bodies were connected to each other above and below the inserted cage/autologous tricortical pelvic bone grafts for stability purposes.

### 2.3. Hemoglobin (Hb) Control of the Patient Population Pre- and Post-Operative

In order to find out whether one of the two surgical interventions resulted in greater bleeding complications in the patient population, the hemoglobin (Hb) values (g/dL) were checked both preoperatively and on the first postoperative day and the difference (ΔHb) was determined.

### 2.4. Statistics

Statistical analysis was performed using the statistic software R (version 3.6.1 for Windows; R Core Team 2018) using the R-package ordinal (version 2019.4.25, Lawrence Livermore National Security, Livermore, USA) for the cumulative link models, and the R-packages emmeans (version 2019.4.25) and ggeffects (version 2019.4.25) for the textual reporting of the statistical models. D’Agostino-Pearson test was used to check for normal distribution. The significance calculation was based on the unpaired *t*-test and the significance level was set to alpha = 5%.

The data have been summarized for all included patients as well separately for the two intervention groups by mean ± standard deviation and median (minimum; maximum) or by absolute and relative frequencies as appropriate.

Multiple models based on the type of intervention, age, and localisation were fit to bony fusion (logistic regression), length of stay (linear regression), and the hemoglobin difference (linear regression). Resulting model coefficients were reported with corresponding 95% confidence intervals and *p*-values against the Null-hypothesis of no association.

Due to the exploratory character of the study no adjustment for multiple testing was applied and *p* values are reported.

## 3. Results

A total of 103 patients treated with cage implantations are in line with the inclusion criteria, but only 12 patients (11.65%; 5 females and 7 males) had complete medical records over a period of at least one year after the surgery, and only those were therefore included in the study. Thirty-four patients (35.79%; 11 females and 23 males) out of the 95 patients with autologous tricortical pelvic bone grafts were included in the study. The gender difference in the cage group (*p* = 0.1969) and in the autologous pelvic bone graft group (*p* = 0.2863) was not statistically significant. Patient material of both groups were examined for the bony fusion of stabilized vertebral segments. The demographic and clinical data of the patients are summarized in Table 1. The logistic model performed to predict bony fusion with type of intervention, age, and localisation showed no significant differences.

The average age of the patients with cage implantation and autologous pelvic bone grafts was different (*p* = 0.0626), i.e., the patients treated with autologous pelvic bone grafts were younger (mean age: 42.18 years) than those with cage implants (mean age: 54.08 years). Of the 12 patients treated with cage implantation, 9 patients showed bone fusion (75%) with no sign of loosening bone or implant during the average follow-up time period of 18.08 months (see Figure 2a). Their mean stationary stay in the clinic was 17.50 days (see Table 1). The mean value of the postoperative inpatient course after the last operation treatment was 6.5 days (±2.84). Of the 34 patients treated with an autologous pelvic bone graft, 18 patients showed bone fusion (52.94%) with no sign of loosening bone or implant during the average follow-up time period of 13.11 months (see Figure 2a). Their mean hospital stay was 23.85 days, i.e., on average almost one week longer than the patients with cage implants, which was statistically significant (*p* = 0.0089) (see Table 1). In addition, the postoperative inpatient stay after the last surgical intervention was on average approx. 6.32 days longer (12.82 days ± 9.11), which was also statistically significant (*p* = 0.0070).

The linear model performed to predict length of stay with type of intervention, age, and localisation showed a significant difference for age (*p* < 0.001). Furthermore, the Fisher’s exact test show a significant difference (*p* = 0.0216) of the bony fusion of the patient group treated with cage implantation and autologous pelvic bone graft, i.e., the cage implants showed a better bony fusion rate than the autologous pelvic bone grafts as determined by a complete bony fusion in the absence of signs for bone loosening. Based on this judgement [11], the minimum fusion time was estimated to be 3.3 months (±3.2) on average in the autologous pelvic bone grafts group and about 4.1 months (±2.8) in the cage group (see Figure 2b). The maximum fusion time observed was lower in the autologous pelvic bone grafts group with an average of 13 months (±16) and in the cage with 9.9 months (±5.2) (see Figure 2c). In addition, the length of the post-surgery hospitalization after cage implantation (mean 17.50 days ± 9.89) and autologous pelvic bone graft (mean 23.85 days ± 17.24) varied in a sense that patients with cage implantations were shorter in inpatient treatment than those with autologous pelvic bone graft. Obviously, these observations show only a tendency because the difference is not statistically significant (*p* = 0.0895).

Most notably, there was a significant gender difference in the fusion duration. Women showed fusion rates of 72.73% (*n* = 11) after autologous pelvic bone graft whereas the fusion rate of men was only 43.48% (*n* = 23). After cage implantation, women showed a fusion rate of 60% and men 85.71%. This gender difference in the fusion duration of cage implantation and autologous pelvic bone graft between men and women was confirmed by the Chi-square test, showing a high significance (*p* < 0.0001) in both cases with regard to the more effective fusion rate of male as compared to female patients.

The hemoglobin (Hb) control of the patients with cage implantations showed an average preoperative value of 12.60 g/dL ± 2.60 and postoperatively 11.00 g/dL ± 2.20, respectively, indicating an average Hb decrease (ΔHb) of 2.39 g/dL ± 1.60. The Hb control of the patient group with autologous pelvic bone graft shows a similar trend with an average preoperative value of 13.09 g/dL ± 2.01, postoperatively 10.58 g/dL ± 1.77 and an average Hb decrease (ΔHb) of 2.39 g/dL ± 1.29 (see Figure 2d). These data indicate that there was no significant difference between the two operative interventions (*p* = 0.0738). The linear regression performed to predict hemoglobin difference with type of intervention, age, and localisation showed no significant differences. It indicates that none of the two surgical interventions resulted in greater bleeding complications.

The bony fusion of patients with cage implants was examined by visual inspection of X-ray images of the implantation site between 2 days and up to 72 months after the surgery. If no signs for loosening were observed, fusion was assumed to be completed three to six months after the implantation. The radiological follow-up was far apart, as the patients were admitted after 3 and 12 months, but these appointments were in fact performed at a much later date. For the cages, the latest appointment was held after 32 months and for the group with autologous pelvic bone graft after 72 months. Complete bone fusion at the implantation site, i.e., the absence of any loosening signs, was observed in 100% of the cases in the area of the thoracal spine 7/8 (Th7/8), the lumbar vertebrae 4/5 (L4/5) and the thoracolumbar junction (Th12/L1). However, in the lower thoracic region, i.e., at Th9/10 (*n* = 2) and Th11/12 (*n* = 4), complete bone fusion was observed in only 50% and 66.67% of the patients, and to about 50% in the area of the lumbar spine, i.e., at L1/2 (*n* = 2) and L3/4 (*n* = 1), respectively (see Table 2). The remaining patients of the corresponding heights still showed signs of loosening. It must be noted that these values provide only a tendency, since meaningful statistical values cannot be obtained in view of the low number of patients.

The bony fusions after autologous pelvic bone grafts (see Table 3) showed similar results as cage implants. At Th4/5 (*n* = 1) and Th10/11 (*n* = 1), 100% complete bone fusion was observed. At L3/4 (*n* = 5) 80%, followed by a fusion rate at L1/2 (*n* = 3) of 66.67% and 50% at Th12/L1 (*n* = 14). Only 25% bony fusions were found to be completed at Th11/12 (*n* = 4) and L2/3 (*n* = 4), and no bony fusion was observed at the following heights of Th5/6 (*n* = 1) and Th7/8 (*n* = 1). The number of revisions that took place after incomplete bone fusion was also very similar among the patients of the two groups. Out of a total of three cage implantation patients that lack complete bone fusions, one required surgical revision (33.33%, 1/3). Five of 16 loose implants (31.25%) of patients with autologous pelvic bone grafts needed a second surgery.

## 4. Discussion

Cage implantations and autologous pelvic bone grafts have been effective standard methods for adjusting spine instabilities in order to restore the proper anatomical positioning of an injured spine [12,13]. The transplantation of bone or bone substitute material has been supposed to lead to better healing of bone defects and a higher degree of stabilization [7]. However, our data do not confirm this earlier study, since our study shows that patients with cage implantations had a significantly better bony fusion than those which received autologous bone grafts. Furthermore, the number of treatment failures was also significantly higher in this group (47.06%) than in the group after cage implantation (25%).

A disadvantage of bone grafts is also that an additional surgical trauma is generated at the donor site, mostly the pelvic bone, which carries an additional surgical risk [7]. Furthermore, enhanced fracture risk in the area of the donor site on the iliac crest as well as hematomas, seromas, and wound infections are known among the complications requiring revision [7]. A high percentage of patients also complained of severe pain at the donor site, which persisted for several weeks [7]. This could also be the reason for the longer inpatient stay of these patients, which was on average 23.85 days. In contrast, patients which received a cage implantation were on average only about 17.50 days hospitalized, which was statistically significant (*p* = 0.0089).

The two surgical interventions compared in the present study were similar with respect to disc removal and surgical access from the ventral side to refill the disc space with a cage insertion or a bone graft. Thus, the differences observed cannot be attributed to the surgical technique applied but rather to the implanted materials. In this context it is interesting to note that our study revealed significant differences (*p* = 0.0216) in the bony fusion between cage implantations and autologous pelvic bone grafts with regard to a bony fusion in the absence of signs for bone loosening. Although the autologous pelvic bone graft showed a significantly higher bony fusion in women, the number of patients with *n* = 11 was significantly lower than that of the men with *n* = 23 (*p* < 0.0001) and the fusion duration of cage implantation showed a significantly higher bony fusion in men (*p* < 0.0001). The cage implantation itself showed a significantly longer bony fusion time (*p* = 0.0029) than autologous pelvic bone grafts. This difference could be attributed to irregular follow-up examinations of patients, which were variable in time. Thus, the fusion rate could not be determined in a highly standardized manner and the corresponding data represent the best possible approximation. Another reason for the better results with cage implantation could be that cages, in contrast to the body-own bony material, cannot be resorbed after the surgery. Furthermore, unlike autologous pelvic bone graft, cage implants are precisely matched to the dimension of the removed vertebral of patients, providing a stable contact surface. Notably, autologous pelvic bone grafts do not have such an optimal fit and also lack the stability as provided by the cages because of the softer consistency.

The fact that the cage implantations are more successful in men than in women could result from a reduced bone quality due to osteoporosis, since the group of women includes postmenopausal patients. Elderly people of both sexes, in particular especially postmenopausal women, have a higher fracture risk due to osteoporosis. Osteoporosis is caused by ovarian hormone deficiency in postmenopausal women [14] and often diagnosed after ovariectomy of premenopausal women [15], suggesting that reduced ovarian hormone levels participate in the cause of the disease. This could be a reason for a better bone quality in men.

The success rate of bony fusion is distinctly different in various regions of the spine. It correlates directly with the different biomechanical aspects of the respective regions. The cervical spine, for example, shows the most extensive range of motion with the highest mobility in segments 5/6 and 6/7 [16,17]. Furthermore, high mobility is also found in the lower thoracic spine in the regions of the thoracolumbar junction and the lower lumbar spine. Those regions showed correspondingly high bony fusions, whereas fusions in the more rigid upper thoracic spine regions were significantly lower.

It has been reported previously that movement is indeed an important factor for better bone healing [18]. The structure and composition of the bones as well as their stability depend essentially on the extend bones being exposed to mechanical stress, which forces act on the bone architectures to enhance the process of bone remodeling, i.e., higher mobility of spinal segments provokes better and faster bone healing [18]. Conversely, the spine areas with high mobility also represent the vulnerable zones of the spine, in particular those regions where mobile elements are adjacent to more rigid sections. Those sections represent degeneration-sensitive areas which are in fact at the transitions of the joint region at the two uppermost cervical vertebrae supporting the head, the cervical spine and thoracic spine, the thoracic and lumbar spine, and the lumbar spine L5 and sacrum S1. The positive correlation between mobility, degeneration-sensitive sites and healing efficiency is indeed reflected in our datasets. It turns out that the most injury-affected region of the spine had been the lumbar spine area, where 41.67% of the patients received cage implantations, mainly in the area of the thoracolumbar transition, i.e., the region where the rigid thoracic spine reaches into the movable lumbar spine. The rest of the cages were inserted to fix the area of the lower thoracic spine at the junction of the thoracic spine. Similarly, the autologous pelvic bone grafts were also primarily applied to replace the discs in the lumbar spine area, 35.29% in the area of the lower thoracic spine, and at the site of the thoracolumbar junction (41.18%). In the thoracic spine, the number of grafts were 23.53%, predominantly in the area of the middle and lower thoracic spine.

A comparison of the duration of the stationary stay in the hospital indicated that the patients with cage implantation were hospitalized for a shorter period than those receiving autologous pelvic bone graft (*p* = 0.0895). This phenomenon is likely to be due to the additional surgical trauma generated at the bone donor site. This conclusion is based on the fact that a high proportion of these patients complain about severe pain at the donor site, which may persist for several weeks. In addition, further complications may have occurred such as hematomas, seromas, or wound infections and fractures in the area of the donor site on the iliac crest, which are generally known among the complications requiring revision.

A significant difference in the postsurgical bony fusion was observed with respect to the gender of the patients. Women showed in autologous pelvic bone graft fusion rates of 72.73% as compared to the 43.48% observed with men (*p* < 0.0001). In contrast, women showed a bony fusion of 60% and men of 85.71% (*p* = 0.0001) after cage implantations. These differences can be attributed to the age distribution in the patient groups. Whereas the average age of the patients that received cage implantations was about 54 years, the average age of patients with autologous pelvic bone grafts was only about 40 years. Thus, the fraction of women that have reached the postmenopausal age, which is often associated with osteoporosis [15], was higher in the group of patients which received cage implantations. It is therefore likely that the bone quality has more osteoporotic character in case of patients with cage implantations. Osteoporosis is characterized by a defective remodeling of the bone substance and a pathological microarchitecture of the bones. Since this disease is left undetected up until the first fracture occurs, it is likely that the difference between bone regeneration after the surgery of women and men is due to this unknown aspect of our study.

Our study also has still some limitations. Since the standard classifications regarding X-ray criteria of loosening are not yet defined, it is suggested that the subjective viewpoints of the surgeon and radiologist played an important role. No computer tomography (CT) diagnostics were performed in patients who were free of symptoms and lacked signs of loosening, although CT diagnostics are more sensitive to determine fusion [10]. In general, therefore, it is difficult to determine fusion, as there is still no uniform classification that can determine it accurately. Overall, the number of patients included in this study appears to be rather small. The reason for this is that fusion surgery of the ventral spine is mainly performed with patients suffering from degenerative diseases, whereas patients with traumatic vertebral body fractures often do not need a ventral fusion. This is due to the very selected patient clientele, which guarantees a best possible comparison. Furthermore, surgeries were carried out at different times. Ideally, patients should have been recruited with a similar health status and nearly identical injuries to ensure identical parameters for an even better comparability. In practical clinical terms, however, such a situation for a perfect comparison cannot be generated. In addition, the number of patients who receive monosegmental ventral spondylodesis of the thoracic and lumbar spine after a vertebral body fracture is generally rather small, in general this fusion surgery of the ventral spine is mainly performed with patients suffering from degenerative diseases. In addition, the exact time point of fusion cannot be determined, because of the incompliance of some of the patients who did not arrive in time for the follow-up appointment. In these cases, the time of fusion could only be approximated. Furthermore, confounding variables such as smoking, diabetes, and osteoporosis were not examined between the two groups, which of course may also have influenced the results. Despite these limitations, the operative treatment of patients with cage implantation and autologous pelvic bone grafts show that the bone fusion tends to be better after cage implantation. Importantly, patients with cage implantations had a significantly shorter stationary stay than patients with autologous pelvic bone grafts. These findings support the argument that cage implantations should be the method of choice if for medical reasons both surgical methods could be applied.

## 5. Conclusions

Both cage implantations and autologous pelvic bone grafts allow direct and individual readjustments of the defective spine region. The timing of subsequent bony fusions depends on the site of the treatment and correlates with the mobility of the corresponding spine region. It appears that sites with high mobility of spinal segments provoke better and faster bone healing, irrespective of the implantation method applied. Our study also revealed that the time for a complete bony fusion in both patients groups is age-dependent and gender-specific. Notably, cage implantations result in a significantly shorter hospitalization period after the surgery. This finding provides an economic argument against autologous pelvic bone grafts. Finally, cage insertions have the advantage that no possible additional surgical trauma is generated at the pelvic donor site. In summary, these results argue for the use of cage transplantations instead of autologous pelvic bone grafts when both methods would be applicable from a medical standpoint.

## Figures and Tables

**Figure 1 medicina-57-00786-f001:**
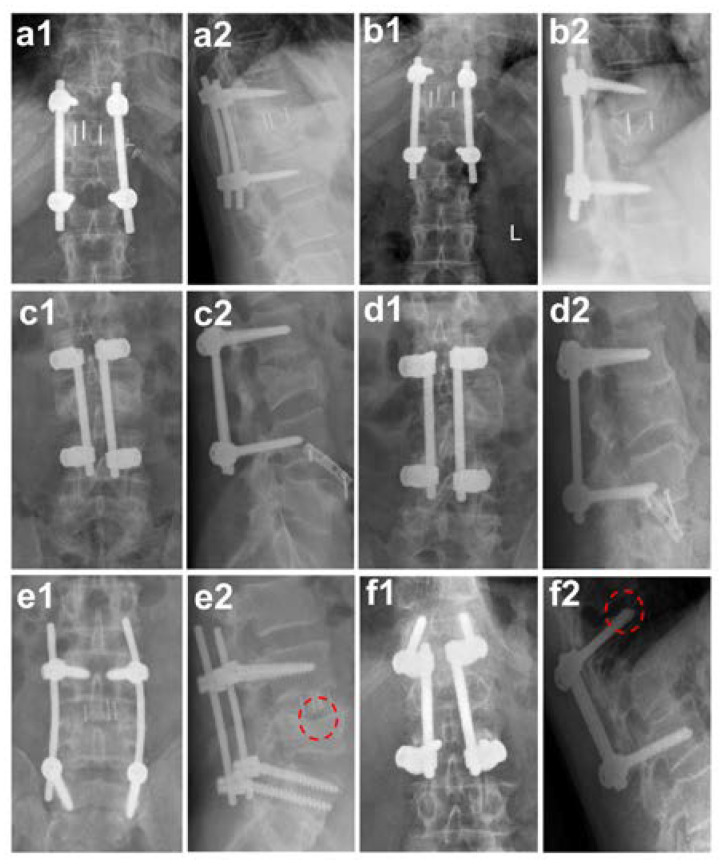
Fusion and non-fusion after surgical care. (**a**,**b**) Cage implantation; (**c**,**d**) autologous pelvic bone graft and subsequent bony fusion; (**a1**,**b1**) spine radiologically in anterior-posterior projection at different time points ((**a1**) post-operative; (**b1**) follow-up after one year); (**a2**,**b2**) spine radiologically in lateral projection at different time points ((**a2**) post-operative; (**b2**) follow-up after one year); (**c1**,**d1**) spine radiologically in anterior-posterior projection at different time points ((**c1**) post-operative; (**d1**) follow-up after one year); (**c2**,**d2**) spine radiologically in lateral projection at different time points ((**c2**) post-operative; (**d2**) follow-up after one year); (**e**,**f**) are controls after one year post-operative and show examples of non-fusion: (**e**) shows a hypodense area around the implanted cage ((**e1**) anterior-posterior projection and (**e2**) lateral projection; see hypodense area marked with dashed circle in red); (**f**) shows the loosening and cutting-out of the screws in the area of the cover plate of the vertebral body after autologous pelvic bone graft insertion ((**f1**) anterior-posterior projection and (**f2**) lateral projection; see cutting-out of the screws marked with dashed circle in red).

**Figure 2 medicina-57-00786-f002:**
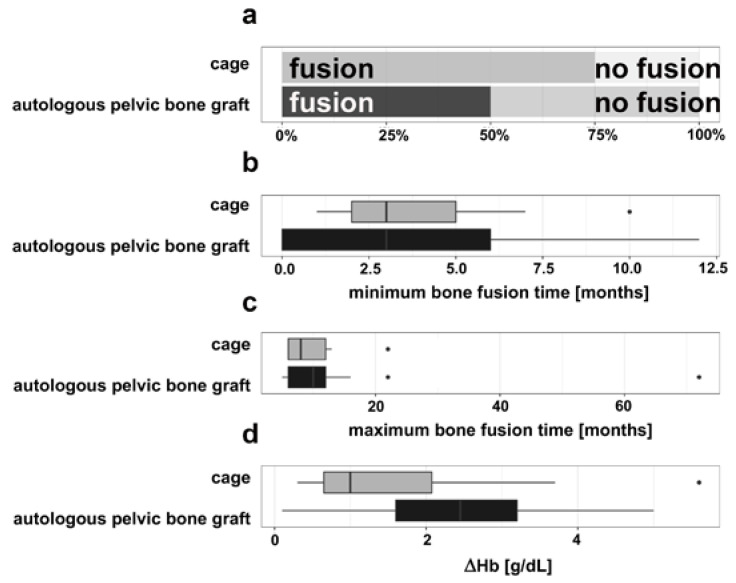
(**a**) Bony fusion of cage implantation (*n* = 12) and autologous pelvic bone graft (*n* = 34). *p* = 0.0216. (**b**) Minimum bone fusion time. Shown data as median values with minimum and maximum. The isolated point represents a data outlier. (**c**) Maximum bone fusion time. Shown data as median values with minimum and maximum. The isolated points represent data outliers. (**d**) Average Hb decrease (ΔHb) between an average preoperative value and an average postoperatively value of the cage implantation collective (*n* = 12) and the autologous pelvic bone graft collective (*n* = 34). Shown data as median values with minimum and maximum. The isolated point represents a data outlier.

**Table 1 medicina-57-00786-t001:** Baseline characteristics of the population.

	Cage	Autologous Pelvic Bone Graft	*p*-Value
Number of patients	12	34	
Age range (years)	34–75	17–79	
Age mean (years) ± SD	54.08 ± 12.12	42.18 ± 16.98	0.0626
Gender	5 females	11 females	
	7 males	23 males	
Stationary stay range (days)	8–32	9–70	
Stationary stay mean (days) ± SD	17.50 ± 9.89	23.85 ± 17.24	0.0089
Postoperative inpatient stay (days) ± SD	6.5 ± 2.84	12.82 ± 9.11	0.0070
Hb-preoperative mean (g/dL) ± SD	12.60 ± 2.60	13.09 ± 2.01	
Hb-postoperative mean (g/dL) ± SD	11.00 ± 2.20	10.58 ± 1.77	
ΔHb (g/dL)	2.39 ± 1.60	2.39 ± 1.29	0.0738
Fusion duration range (months)	6–22	5–72	
Fusion duration mean (months) ± SD	9.88 ± 5.23	13.29 ± 15.80	
Fusion rate (%)	75.00	52.94	0.0216
Diagnosis (%)			
Fracture/Trauma (%)	100	100	
Location (%)			
Thoracolumbar junction	-	41.18	
Thoracic spine	58.33	23.53	
Lumbar spine	41.67	35.29	

**Table 2 medicina-57-00786-t002:** Bony fusion of cage implantation and different vertebral body heights.

Vertebral Body Height	Fusion (%)
Th7/8 (*n* = 1)	100
Th9/10 (*n* = 2)	50
Th11/12 (*n* = 4)	66.67
Th12/L1 (*n* = 1)	100
L1/2 (*n* = 2)	50
L3/4 (*n* = 1)	50
L4/5 (*n* = 1)	100

**Table 3 medicina-57-00786-t003:** Bony fusion of autologous pelvic bone graft and different vertebral body heights.

Vertebral Body Height	Fusion (%)
Th4/5 (*n* = 1)	100
Th5/6 (*n* = 1)	0
Th7/8 (*n* = 1)	0
Th10/11 (*n* = 1)	100
Th11/12 (*n* = 4)	25
Th12/L1 (*n* = 14)	50
L1/2 (*n* = 3)	66.67
L2/3 (*n* = 4)	25
L3/4 (*n* = 5)	80

## Data Availability

The data that support the findings of this study are available on request from the corresponding author. The data are not publicly available due to privacy or ethical restrictions.

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
