# Peer review of "Cage or Pelvic Graft—Study on Bony Fusion of the Ventral Thoracic and Lumbar Spine in Traumatic Vertebral Fractures"

_medicina, 2021, doi:10.3390/medicina57080786_

Round 1
Reviewer 1 Report
The Introduction sets up the study well. The Materials and Methods appear reasonable. Thank you for including your statistical plan. Results are clear and easy to understand. The discussion is good. Thank you for emphasizing the additional healing time for autologous grafts. Fine paper
Reviewer 2 Report
Thank the authors for revising the manuscript in response to the reviewers' comments. The comments have been adequately addressed.
This manuscript is a resubmission of an earlier submission. The following is a list of the peer review reports and author responses from that submission.
Round 1
Reviewer 1 Report
Dear authors, I read your article with interest. You report your results comparing pelvic bone autograft to allograft cages in stabilizing Thoraco Lumbar trauma. Surprisingly you find that allograft has better results than autograft in terms of achieving fusion. These findings are counterintuitive to the well established fact that autograft is superior to allograft. As you yourself point out in the manuscript, technical differences may be the reason that, in your series, allograft provided better results. Also, confounding variables like smoking, diabetes, osteoporosis, and so on, also have not been studied between the groups.
Reviewer 2 Report
Abstract is nicely structured and organized.
Introduction is nicely written and provides enough background for the study and also explains why the study is of importance. The background and literature review is appropriate and sets up the study of synthetic cage implants versus autologous pelvic bone grafts.
Material and Methods
Thank you for including your ethics statement. Inclusion/exclusion criteria is clear.
Results
Results are organized nicely. Table one is easily understandable and displays the demographic data well. Statistical results follow the stated statistical plan from materials and methods in a logical fashion. Makes the paper easy to read.
Discussion
Thank you for including the additional trauma from the donor site. I am also inclined to believe that the longer inpatient stay is linked to this as well. Limitations for on evaluating fusion are nicely stated. However, for concerns over exposing patients to too much radiation, it is unreasonable to expect every patient to receive a CT scan and radiographs should be expected.
This is a very sound and nicely written paper. It was difficult to review as there was not a lot wrong with it. I commend the authors on a nice, clean and well written paper.
Reviewer 3 Report
This paper compares surgery outcomes from the use of autologous (pelvic graft) and synthetic (cage) implants for spinal fusion. The study deals with an interesting topic, but the conclusion drawn from this study was not significant for several reasons. First, the bony fusion was evaluated by only a single method, X-ray imaging, which is not quantifiable and can be easily biased. Second, this evaluation was also done at non-identical time points. Lastly, the patient group size was too small to make meaningful conclusions in some portion of the study. The below is some suggestions that could strengthen the study and the manuscript.
- The authors took into account the osteoporosis when interpreting the results from pelvic graft bone implants. In addition to the mechanical property, biochemical and cellular characteristics of the pelvic bone implant can considerably affect the quality of the autologous implants and their performance. Therefore, further analysis of cytokines and growth factors in the pelvic bone implants would be very interesting.
- Surgical outcomes seem highly dependent on the sex of the patients. Thus, it would be desirable that the sex ratio of the patient keeps the same between the two implant groups.
- The quantifiable scoring system of bony fusion and healing outcome would be helpful for systematic investigation.
- The inclusion of typical X-ray image examples of successful and failed infusion could be informative.